# Dietary Habits and Dietary Antioxidant Intake Are Related to Socioeconomic Status in Polish Adults: A Nationwide Study

**DOI:** 10.3390/nu12020518

**Published:** 2020-02-18

**Authors:** Małgorzata Elżbieta Zujko, Anna Waśkiewicz, Wojciech Drygas, Alicja Cicha-Mikołajczyk, Kinga Zujko, Danuta Szcześniewska, Krystyna Kozakiewicz, Anna Maria Witkowska

**Affiliations:** 1Department of Food Biotechnology, Faculty of Health Sciences, Medical University of Bialystok, Szpitalna 37, 15-295 Bialystok, Poland; z_kinga@wp.pl (K.Z.); witam@umb.edu.pl (A.M.W.); 2Department of Epidemiology, Cardiovascular Disease Prevention and Health Promotion, National Institute of Cardiology, Alpejska 42, 04-628 Warsaw, Poland; awaskiewicz@ikard.pl (A.W.); wdrygas@ikard.pl (W.D.); acicha@ikard.pl (A.C.-M.); dszczesniewska@ikard.pl (D.S.); 3Department of Social and Preventive Medicine, Faculty of Health Sciences, Medical University of Lodz, Hallera 1, 90-001 Lodz, Poland; 4Division of Cardiology and Structural Heart Diseases, Medical University of Silesia, Ziolowa 45, 40-635 Katowice, Poland; kozakiewicz.krystyna@gmail.com

**Keywords:** dietary habits, dietary antioxidants, socioeconomic status, population

## Abstract

The aim of this study was to estimate dietary habits and dietary antioxidant intake in a Polish adult population in relation to socioeconomic status. The subjects (4774) were participants in the Polish National Multi-Centre Health Examination Survey (the WOBASZ II study) performed in 2013–2014. Socioeconomic status (SES) scores were calculated by multiplying ordinal numerical values assigned to consecutive categories of education level and monthly income per capita in a family. In the Polish adult population, a higher socioeconomic status was significantly associated with a better lifestyle (more physical activity and less smoking), a better health status (lower occurrence of overweight individuals and metabolic syndrome in both genders, and lower occurrence of central obesity, hypertension, and diabetes in women), and better dietary habits, including a higher intake of dietary antioxidants.

## 1. Introduction

Noncommunicable diseases, such as cardiovascular diseases, cancer, diabetes, and chronic lung disease, are responsible for ~70% of all deaths worldwide. Therefore, the prevention and treatment of these diseases are fundamental public health concerns. Some risk factors, including age, ethnic background, and genetic determinants, are classified as non-modifiable. Of the modifiable risk factors, tobacco use, physical inactivity, the harmful use of alcohol, and an unhealthy diet are the most important [1]. Epidemiological studies have established an inverse association between a healthy diet that is rich in antioxidants (antioxidant vitamins and polyphenols) and the risk of cancer [2,3], metabolic disorders [4], and cardiovascular diseases [5,6].

Dietary antioxidants, such as polyphenols, antioxidant vitamins (vitamins C, E, and A), and minerals (zinc, iron, copper, manganese, and selenium), which are components of antioxidant enzymes, help the internal antioxidant system to reduce oxidative stress, which is involved in the pathogenesis of chronic diseases. Polyphenols form the most abundant group of antioxidants and are found in commonly consumed plant-based foods. The average polyphenol intake in different populations is estimated to be 1–2 g/day. Dietary antioxidant intake is dependent on the polyphenol, vitamin, and mineral content in foods (related to genetic, environmental, and processing factors) and individual food preferences [7,8,9].

Dietary patterns are shaped by cultural factors, environmental factors, and socioeconomic status (SES). A lower SES can predispose an individual to the purchase of food of low nutritional density, which can lead to an excess of calorie intake and, consequently, overweight and obese individuals [10]. It was found that adherence to the healthy Mediterranean diet was lower in uneducated individuals, those with a lower income, and those who were unemployed in countries with traditional Mediterranean eating patterns [11]. A study conducted in four European countries (Denmark, France, Italy, and the Czech Republic) found that a large part of the population did not follow food-based dietary guidelines. Moreover, food intakes were found to vary by age, gender, and education level [12]. The results obtained in a previous study concerning the Polish population support the hypothesis that people with a higher socioeconomic status have better access to health care in terms of frequency of medical consultations and receive counselling on nutrition and physical activity [13]. Dietary education should take into account the income of the targeted population and, in low-income individuals, should focus on a healthy and affordable diet [14].

The aim of this study was to estimate dietary habits and dietary antioxidant intake in a Polish adult population in relation to their socioeconomic status. To the best of our knowledge, this is the first attempt to identify relationships between total dietary antioxidant capacity, dietary antioxidant intake (polyphenols, vitamins, and minerals), and socioeconomic status in a cross-sectional Polish study.

## 2. Materials and Methods

### 2.1. Study Population

Subjects were aged 20 years or older and participants in the Polish National Multi-Centre Health Examination Survey (the WOBASZ II study) performed in 2013–2014. From the Polish adult population, a sample of 15,120 individuals was drawn from the Department of State Registry database of the Ministry of Internal Affairs (the PESEL register). Altogether, we recruited 6170 respondents to the study (a response rate of 45.5%). Reliable dietary recalls and information about sociodemographic and health-related factors were obtained from 4774 people (2142 men and 2632 women).

The study protocol was approved by the Bioethics Committee of the National Institute of Cardiology (no. 1344). The aims of and methods used in the WOBASZ II study have been described in detail elsewhere [15,16].

### 2.2. Data Collection

Self-reported data on age, socioeconomic status (education level and monthly income per capita in a family), lifestyle (leisure-time physical activity, smoking status, and alcohol consumption), health status (diabetes and cardiovascular disease), and dietary habits (use of dietary supplements and use of a special diet) were collected from the standardized questionnaires, which were designed for the WOBASZ II study. A low level of leisure-time physical activity was defined as 30 min/day of physical activity at a frequency of once per week or less.

Measurements of body mass, height, waist circumference, and blood pressure were performed by trained nurses using standardized procedures [15]. A diagnosis of hypertension was made if systolic blood pressure (SBP) ≥140 mmHg and/or diastolic blood pressure (DBP) ≥90 mmHg, or a diagnosis of hypertension was self-declared during an interview. Body mass index (BMI) was calculated as body mass in kilograms divided by squared height in meters. A BMI ≥25 was classified as overweight or obese. Central obesity was determined as a waist circumference (WC) ≥102 cm for men and ≥88 cm for women. Measurements of fasting glucose and blood lipids were performed in the Diagnostic Central Laboratory at the Institute of Cardiology in Warsaw, Poland [15]. A diagnosis of diabetes was made if the blood glucose level was ≥7.0 mmol/L, or a diagnosis of diabetes was self-declared during an interview.

Metabolic syndrome (MetS) criteria were adopted in accordance with the International Diabetes Federation (IDF) and American Heart Association/National Heart, Lung, and Blood Institute (AHA/NHLBI) definition [17]. A diagnosis of MetS was made when at least three of five risk factors were identified: (1) an elevated WC (in the European population: ≥94 cm for men and ≥80 cm for women); (2) an elevated triglyceride (TG) level (≥150 mg/dl (1.7 mmol/L)); (3) a reduced high-density lipoprotein cholesterol (HDL-C) level (<40 mg/dl (1.0 mmol/L) for men and <50 (1.3 mmol/L) for women); (4) elevated blood pressure (SBP ≥130 mm Hg and/or DBP ≥85 mm Hg) or self-declared hypertension in an interview; and (5) an elevated fasting glucose (FG) level (≥100 mg/dl (5.6 mmol/L)) or self-declared diabetes in an interview.

### 2.3. Nutrition Assessment

A single 24-h dietary recall was used to assess dietary food product intake. Food portion sizes were estimated using an album with photographs of the most-consumed food products [18]. Food Composition Tables were used to assess the nutritional value of the daily food rations [19]. The healthy diet indicator (HDI) score was calculated using seven components. According to the WHO recommendations, the following cut-off value were adopted: saturated fatty acids <10% TE (total energy without energy from alcohol), free sugars <10% TE, polyunsaturated fatty acids 6–10% TE, protein 10–15% TE, cholesterol <300 mg/day, fibre ≥25 g/day, and fruits and vegetables ≥400 g/day. The HDI score ranged from 0 (the least healthy diet) to 7 (the healthiest diet) and was classified as follows: high (6–7 points), moderate (4–5 points), and low (0–3 points). We omitted the salt component, because the amount of added salt during preparation of meals and at the table could not be evaluated precisely. However, adding salt to already seasoned dishes was reported by ¼ of Polish adults [16,20].

Dietary antioxidant capacity and dietary polyphenol intake were determined by multiplying the daily consumption of individual food items by the antioxidant capacity (measured by the ferric reducing antioxidant potential (FRAP) method) of these food items and the polyphenol content (measured by a Folin–Ciocalteu assay) in these food items, respectively, using the databases [21,22,23,24].

### 2.4. Socioeconomic Status Scores

Socioeconomic status (SES) scores were calculated by multiplying ordinal numerical values assigned to consecutive categories of education level and monthly income per capita in a family [25,26]. Education level was divided into nine groups, and monthly income (net) per capita in a family was divided into seven groups (Table 1). The SES score ranged from 1 to 63. According to tertile values of the SES score distribution, study participants were divided into three groups: low (1–8), medium (9–18), and high (19–63) SES scores.

### 2.5. Statistical Analysis

Descriptive statistics were used to describe continuous variables, and the percentages of the respective values were used to describe categorized variables. A chi-squared test was used to compare the distribution of education level and income level between men and women (Table 1).

An analysis of covariance (GLM proc) with an age adjustment was applied to determine frequencies (risk factors and incidence of diseases) and means (nutritional factors) in tertiles of socioeconomic status (Table 2, Table 3 and Table 4).

A Kruskal–Wallis test was used to compare the main food sources of dietary antioxidant capacity (Table 5).

The level of significance for bilateral tests was set at *p* < 0.05. Statistical analyses were performed using the Statistical Analysis System (SAS) version 9.2.2 software (SAS Institute Inc., Cary, NC, USA).

## 3. Results

General characteristics of the studied population according to the SES score are presented in Table 2. In men (*N* = 2142), the SES score was low (L–SES) for 740 participants, medium (M–SES) for 745 participants, and high (H–SES) for 657 participants. In women (*N* = 2632), the L–SES group had 913 participants, the M–SES group had 918 participants, and the H–SES group had 801 participants. It was found that the SES score was significantly (*p* < 0.0001) associated with the age of the studied population; therefore, in further analyses, the data were adjusted for age. An analysis of the lifestyle and health status in the SES groups showed that both men and women with H–SES were less likely to have a low level of physical activity, be a current smoker, overweight, or obese, and have a metabolic syndrome, and women with H–SES were less likely to have central obesity, hypertension, and diabetes.

The dietary habits of the studied population (*N* = 4774), according to the SES scores and adjusted for age, are presented in Table 3. No differences were found between the SES score categories and the healthy diet indicator. The HDI was relatively low in each SES group and ranged between 3.18 and 3.29. Regardless of the SES group, dietary habits of the Polish adults are below the recommended standards, as we have shown in a previously published paper [16]. The traditional Polish diet is characterized by a high intake of red meat and animal fat, and a low intake of fish, pulses, whole grain cereal products, vegetables, seeds, and nuts. Among vegetables, the most popular are potatoes, cabbage, and beetroot. The most popular fruits are apples. Red wine and spices—rich in antioxidants—are not typical in a Polish diet. Among alcoholic beverages, Polish adults most frequently drink vodka and beer, which are poor in antioxidants. However, higher SES groups consumed products with higher content of polyphenols, antioxidant vitamins, and minerals. These components are not included in the HDI score. Moreover, participants with a higher SES frequently used dietary supplements and a weight-loss diet. Other diets, such as low-fat, low-cholesterol, and diabetic diets, were mainly used by subjects with a lower SES. A higher SES was positively associated with better dietary practices in terms of a lower energy intake and a higher intake of wholemeal bread, vegetables, and dairy products in both genders, a lower intake of red meat in men, and a higher intake fruits, fish, and nuts in women. The analysis of beverage consumption showed that men with H–SES consumed less tea and more sugar-sweetened beverages and fruit and vegetable juices; however, women with H–SES consumed less tea and more coffee. The consumption of plant fats (oil and margarine) and cereal products was associated with a lower SES in both genders. Legume intake was found to be associated with a lower SES only in men.

Dietary antioxidant capacity and dietary antioxidant intake were calculated based on 1000 kcal and are presented in Table 4. The results showed that dietary antioxidant capacity, dietary polyphenol intake, and intake of antioxidant vitamins (C, E, A) and minerals (zinc, iron, copper, and manganese), including supplements, were higher in participants with a higher SES. These relationships were not found only for vitamin A in the male group.

The main food sources of total dietary antioxidant capacity were found to be coffee, tea, vegetables, fruits, and, in smaller quantities, cereals, nuts and seeds, and chocolate and cocoa (Table 5). Men with H–SES consumed significantly more dietary antioxidants from nuts and seeds and chocolate and cocoa, and less dietary antioxidants from tea and cereals in comparison to the lower SES groups. In women, a higher SES was significantly associated with a higher intake of dietary antioxidants from coffee, nuts and seeds, and chocolate and cocoa, and a lower intake of dietary antioxidants from tea and vegetables.

## 4. Discussion

In this study, we found a significant association in a Polish adult population between socioeconomic status and lifestyle, health status, and dietary habits, including dietary antioxidant intake. The results of the present study have implications for the design of future nutrition promotional strategies for the Polish population.

Polish adults with a higher SES were less predisposed to smoking, a low level of leisure-time physical activity, being overweight, and having a metabolic syndrome in both genders and central obesity, hypertension, and diabetes in women. This finding is consistent with the results of other studies. It is well-documented that a low socioeconomic status (education level and income) has a significant impact on the incidence of obesity, diabetes, and cardiovascular disease [27,28,29]. In the cross-sectional EUROASPIRE IV study, which was undertaken in 24 European countries, a higher level of education was found to have a significant impact on the control of risk factors in patients with coronary heart disease, such as currently smoking, being overweight, obesity, a low level of physical activity, low HDL-C, hypertension, and diabetes [30]. These relationships can be explained by differential access to medical care and behavioural and psychosocial factors. Individuals with a lower socioeconomic status are more likely to engage in poor health-related behaviours, such as smoking, an unhealthy diet, and a sedentary lifestyle. A nationally representative survey from Korea showed that adolescents with a low level of household income, whose fathers had a low level of education, were more likely to have an early initiation into smoking and drinking alcohol [31]. A study carried out in Poland in a working-age population found that respondents with a steady and higher income had the highest physical activity levels [32]. A cross-sectional analysis conducted in Germany with nearly 20,000 adults demonstrated that multimorbidity, defined as the presence of more than one chronic disease at the same time in one individual, was associated with the age and education of participants [33].

The results of a meta-analysis showed that, in developed countries, women with a higher socioeconomic status throughout their life have a lower BMI, while the findings in men were less consistent. These results can be explained by the fact that women may more often that men have weight-related ideals that are easier to maintain with a higher income [34]. In our study, women with lower SES were found to be more likely to have central obesity, diabetes, and hypertension; however, these relationships were not found in men.

In this study, participants with a higher SES were found to have better dietary habits than participants with a lower SES. Similarly to the current study, the cross-sectional Australian Health Survey showed that a lower socioeconomic position was associated with a lower-quality diet and lower intake of some foods and nutrients, and these relationships differed by sex [35].

A number of studies have indicated that nutrition plays an important role in the prevention of non-communicable diseases [36,37,38]. Dietary antioxidants, such as polyphenols, vitamins, and minerals, play a special role in the prevention of oxidative-stress-related diseases. A higher intake of polyphenols, particularly flavonoids, has been associated with a lower risk of diabetes, cardiovascular events, and all-cause mortality [39]. Moreover, a higher consumption of dietary antioxidant vitamins reduces the risk of obesity [40] and mortality from cardiovascular disease [41]. A study in the Polish population showed that higher dietary antioxidant capacity and higher polyphenol intake were associated with a lower occurrence of hypertension [42] and diabetes [4].

In the current study, dietary antioxidant capacity, dietary polyphenol intake, and intake of antioxidant vitamins (C, E, and A) and minerals (zinc, iron, copper, and manganese)—including supplements—were found to be higher in participants with a higher SES. Similarly to the current study, a cross-sectional analysis of the PREDIMED-Plus trial in the Spanish population showed that higher nutrition density was directly and significantly associated with a higher education level and better adherence to the Mediterranean diet [43]. The NHANES study, which was conducted in a U.S. diabetes population, found that diet quality was significantly lower in individuals with a lower socioeconomic status, which did not improve over a 16-year period [44].

In this study, the main food sources of dietary antioxidants were found to be coffee, tea, vegetables, fruits, cereals, nuts and seeds, and chocolate and cocoa. These results are in agreement with those of previous studies [7,42]. However, in this study, participants with a higher SES consumed significantly more dietary antioxidants from nuts and seeds and chocolate and cocoa in both genders, and coffee only in women, in comparison to lower SES consumers. In the Polish diet, the consumption of peanuts and sunflower seeds is common, because they are the cheapest. However, our previous study showed that walnuts had the highest antioxidant activity [22]. A previous Polish study showed that individuals who consumed nuts had better dietary habits than individuals who did not consume nuts [45]. Moreover, the Australian Health Survey indicated that habitual consumers of espresso and ground coffee have a higher socioeconomic status than consumers of coffee mixes and instant coffee and non-consumers of coffee [46].

In the current literature, there is a lack of information on dietary antioxidant intake in relation to the socioeconomic status of different populations.

This study exhibits strengths and has limitations. The advantage of this study is that it uses a large group of participants from a representative Polish population and standardized methods. Moreover, it provides data on the relationship between dietary antioxidant intake and socioeconomic status, which to date has not been analyzed in a cross-sectional study. The main limitation of this study is the use of a single 24-h recall, which does not reflect habitual or long-term food intake. However, it is commonly used to estimate the average food intake in a large group of participants. The use of multi-day recalls places more of a burden on respondents and can increase drop-out rates.

## 5. Conclusions

In the present study, a higher socioeconomic status in Polish adults was significantly associated with a better lifestyle (a higher level of physical activity and a lower smoking rate), a better health status (a lower incidence of overweight individuals and a lower occurrence of metabolic syndrome in both genders, and a lower incidence of central obesity, hypertension, and diabetes in women), and better dietary habits, including higher dietary antioxidant intake. The low SES groups should consume food that is relatively inexpensive, but has a high antioxidant potential, such as vegetables (red cabbage, beetroots), legumes (pea, bean) and seasonal Polish fruits (strawberries, raspberries, plums, black berries).

## Figures and Tables

**Table 1 nutrients-12-00518-t001:** Distribution of education level and monthly income (net) per capita in a family, according to gender.

	Men (*N* = 2142)	Women (*N* = 2632)	*p*
**Education level**			
1. Incomplete elementary or uneducated	8 (0.37%)	28 (1.06%)	<0.0001
2. Elementary	313 (14.61%)	485 (18.43%)	
3. Vocational based on elementary school	650 (30.35%)	460 (17.48%)	
4. Gymnasium	21 (0.98%)	9 (0.34%)	
5. Vocational based on middle school	29 (1.35%)	29 (1.10%)	
6. Secondary	670 (31.28%)	776 (29.48%)	
7. Post-secondary	57 (2.66%)	201 (7.64%)	
8. Bachelor’s degree	33 (1.54%)	81 (3.08%)	
9. University	361 (16.85%)	563 (21.39%)	
**Monthly income (net) per capita in a family**			
1. <500 PLN (<125 €)	250 (11.67%)	340 (12.92%)	<0.0001
2. 501–1000 PLN (126–250 €)	658 (30.72%)	932 (35.41%)	
3. 1001–1500 PLN (250–375 €)	515 (24.04%)	713 (27.09%)	
4. 1501–2000 PLN (376–500 €)	344 (16.06%)	343 (13.03%)	
5. 2001–2500 PLN (501–625 €)	161 (7.52%)	155 (5.89%)	
6. 2501–3000 PLN (626–750 €)	92 (4.30%)	77 (2.93%)	
7. >3000 PLN (>750 €)	122 (5.70%)	72 (2.74%)	

Data are shown as ‘number (percentage)’.

**Table 2 nutrients-12-00518-t002:** General characteristics of the studied population (*N* = 4774) according to their socioeconomic status (SES) score (adjusted for age).

	Men *N* = 2142				Women *N* = 2632			
	SES			*p*	SES			*p*
	Low (1–8) *N* = 740	Medium (9–18) *N* = 745	High (19–63) *N* = 657		Low (1–8) *N* = 913	Medium (9–18) *N* = 918	High (19–63) *N* = 801	
Age	54.6 ± 16.0	49.1 ± 15.6	44.3 ± 15.6	<0.0001	58.3 ± 16.4	47.9 ± 15.5	45.9 ± 14.4	<0.0001
Leisure-time physical activity low level	407 (54.7%)	324 (43.7%)	213 (33.0%)	<0.0001	487 (51.2%)	349 (39.0 %)	278 (36.4%)	<0.0001
Smoking status current smokers	260 (37.5%)	225 (30.0%)	135 (18.2 %)	<0.0001	169 (19.9%)	197 (20.9%)	133 (15.7%)	0.0155
BMI (kg/m^2^) BMI ≥ 25	460 (75.7)	520 (72.9)	447 (61.3)	<0.0001	604 (61.9)	525 (63.4)	373 (54.8)	0.0004
Central obesity (≥102 cm—M; ≥88 cm—W)	250 (31.2%)	266 (36.7%)	190 (33.1%)	0.0724	558 (54.4%)	445 (52.7%)	283 (42.0%)	<0.0001
Diseases								
Diabetes	119 (13.8%)	88 (12.4%)	45 (10.3%)	0.1468	152 (13.6%)	72 (9.8%)	48 (8.9%)	0.0050
Hypertension	398 (48.2%)	376 (51.4%)	288 (51.0%)	0.3599	521 (45.7%)	354 (43.8%)	239 (38.3%)	0.0012
Cardiovascular disease	178 (18.9%)	158 (21.6%)	103 (21.1%)	0.3310	266 (22.4%)	163 (20.5%)	139 (21.9%)	0.5524
Metabolic syndrome	303 (36.0%)	326 (44.0%)	233 (39.6%)	0.0151	423 (38.7%)	281 (33.7%)	189 (28.8%)	<0.0001

**Table 3 nutrients-12-00518-t003:** Dietary habits of the studied population (*N* = 4774) according to their socioeconomic status (SES) score, adjusted for age.

	Men				Women			
	SES			*p*	SES			*p*
	Low (1–8)	Medium (9–18)	High (19–63)		Low (1–8)	Medium (9–18)	High (19–63)	
Healthy diet index (points), mean, CI	3.23 3.14–3.33	3.25 3.15–3.34	3.18 3.08–3.29	0.6662	3.26 3.18–3.34	3.26 3.18–3.34	3.29 3.19–3.36	0.9337
Use of dietary supplements (%)	6.52 4.28–8.80	10.30 8.10–10.56	16.86 14.44–19.27	<0.0001	12.76 10.17–15.36	18.41 15.91–20.92	24.65 21.94–27.36	<0.0001
Use of special diet (%)								
Weight-loss diet	0.14	0.27	2.28	<0.0001	0.33	1.31	1.87	<0.0001
Low-fat, low-cholesterol, or diabetic diet	8.38	9.13	4.57		12.81	6.97	5.24	
Other diet	1.35	0.81	2.13		0.44	1.96	3.12	
Energy (kcal/day), mean, CI	2432 2365–2498	2319 2254–2385	2186 2116–2257	<0.0001	1731 1688–1774	1664 1622–1705	1640 1595–1684	0.0129
Cereal products (g/day), mean, CI	202.6 195.9–209.4	190.4 183.7–197.0	164.7 157.5–171.9	<0.0001	135.8 131.4–140.2	127.1 122.9–131.4	120.1 115.5–124.7	<0.0001
Wholemeal bread (g/day), mean, CI	18.8 14.0–23.7	32.9 28.1–37.7	36.2 31.1–41.4	<0.0001	20.9 17.5–24.2	28.5 25.2–31.8	32.9 29.3–36.4	<0.0001
Vegetables (g/day), mean, CI	244.9 230.9–258.9	266.5 252.8–280.3	274.4 259.5–289.3	0.0142	211.7 200.9–222.6	235.1 224.7–245.6	244.4 233.1–255.7	0.0002
Legumes (g/day), mean, CI	5.21 3.92–6.51	3.66 2.39–4.93	2.81 1.44–4.18	0.0420	3.18 2.25–4.09	2.24 1.35–3.12	3.11 2.16–4.07	0.2710
Fruits (g/day), mean, CI	180.6 162.9–198.4	196.4 179.1–213.8	196.3 177.5–215.1	0.3768	196.7 181.1–212.4	222.3 207.2–237.3	235.6 219.3–251.9	0.0036
Red meat (g/day), mean, CI	184.3 172.4–196.2	169.9 158.3–181.6	129.4 116.8–142.1	<0.0001	79.6 73.5–85.8	77.5 71.6–83.5	77.0 70.6–83.5	0.8332
Poultry (g/day), mean, CI	56.4 47.5–65.2	65.1 56.4–73.7	68.6 59.2–78.0	0.1638	56.2 50.1–62.2	51.3 45.5–57.1	46.9 40.6–53.2	0.1268
Fish (g/day), mean, CI	18.3 13.1–23.5	22.7 17.6–27.7	25.9 20.4–31.4	0.1485	12.0 8.4–15.5	12.4 9.0–15.9	18.8 15.1–22.5	0.0158
Dairy products (g/day), mean, CI	367.0 326.2–407.9	416.4 376.4–456.5	541.6 498.2–584.9	<0.0001	383.9 354.0–413.9	441.2 412.3–470.2	474.6 443.3–505.8	0.0003
Animal fats (butter, lard) (g/day), mean, CI	24.7 22.5–26.9	26.3 24.2–28.5	26.5 24.1–28.8	0.4758	19.5 18.1–20.8	20.0 18.7–21.3	19.2 17.7–20.6	0.7030
Plant fats (oil, margarine) (g/day), mean, CI	28.4 26.6–30.3	24.5 22.7–26.4	21.9 20.0–23.9	<0.0001	19.1 18.0–20.3	16.0 14.9–17.1	15.6 14.5–16.9	<0.0001
Tea infusion (mL/day), mean, CI	365.0 344.7–385.2	337.2 317.3–357.1	313.2 291.8–334.8	0.0033	339.5 322.5–356.4	309.6 293.2–326.0	312.5 294.9–330.2	0.0313
Coffee infusion (mL/day), mean, CI	169.7 155.3–184.1	163.6 149.5–177.7	165.6 150.4–180.9	0.8348	179.7 167.3–192.1	193.0 181.0–205.0	210.1 197.1–223.0	0.0050
Nuts and seeds (g/day), mean, CI	1.53 0.48–2.56	2.26 1.24–3.27	3.14 2.04–4.24	0.1227	1.01 0.11–1.91	1.55 0.68–2.41	3.30 2.37–4.23	0.0018
Sugar-sweetened beverages (mL/day), mean, CI	38.0 23.7–52.3	41.7 827.8–55.8	75.5 60.3–90.6	0.0007	20.9 14.2–27.5	16.6 10.2–23.0	11.9 5.0–18.8	0.1992
Fruit and vegetable juices (mL/day), mean, CI	32.8 22.5–43.2	40.3 30.2–50.4	55.7 44.7–66.6	0.0122	34.7 26.8–42.5	44.5 36.9–52.1	44.5 36.4–52.8	0.1460
Alcohol (pure ethanol) (mL/day), mean, CI	4.58 3.32–5.85	4.42 3.17–5.66	4.57 3.22–5.91	0.9801	0.57 0.25–0.90	0.61 0.29–0.92	0.74 0.40–1.08	0.7618

CI—Confidence Interval.

**Table 4 nutrients-12-00518-t004:** Dietary antioxidants intake according to socioeconomic status (SES) score, adjusted for age.

	Men				Women			
	SES			*p*	SES			*p*
	Low (1–8)	Medium (9–18)	High (19–63)		Low (1–8)	Medium (9–18)	High (19–63)	
Dietary antioxidant capacity (mmol/day), mean, CI	12.46 11.90–13.03	12.59 12.03–13.14	12.37 11.77–12.97	0.8665	11.59 11.09–12.09	12.34 11.85–12.82	13.15 12.63–13.67	0.0002
Dietary antioxidant capacity/1000 kcal (mmol/day), mean, CI	5.44 5.19–5.69	5.77 5.53–6.02	6.15 5.89–6.42	0.0010	7.26 6.92–7.59	7.95 7.63–8.28	8.49 8.14–8.84	<0.0001
Dietary polyphenol intake (mg), mean, CI	2120.9 2051–2190	2090.5 2022–2159	2038.6 1964–2113	0.2931	1923.7 1866–1981	1985.7 1931–2041	2069.3 2010–2129	0.0031
Dietary polyphenol intake/1000 kcal (mg), mean, CI	917.6 886.7–948.5	958.9 886.7–989.2	1005.5 972.7–1038.3	0.0009	1188.4 1150–1227	1271.3 1234–1309	1335.4 1295–1375	<0.0001
Vitamin C with supplementation (mg), mean, CI	77.8 71.8–83.7	85.6 79.8–91.5	100.9 94.6–107.2	<0.0001	78.2 72.0–84.4	97.2 91.3–103.2	109.5 103.1–116.0	<0.0001
Vitamin C with supplementation/1000 kcal (mg), mean, CI	34.7 31.7–37.8	40.0 37.0–43.0	50.1 46.8–53.3	<0.0001	47.8 43.4–52.2	63.1 58.8–67.3	72.9 68.4–77.6	<0.0001
Vitamin E with supplementation (mg), mean, CI	12.5 11.9–13.1	12.8 12.2–13.4	12.6 12.0–13.3	0.8227	10.3 9.3–11.3	10.5 9.5–11.4	12.2 11.1–13.2	0.0211
Vitamin E with supplementation/1000 kcal (mg), mean, CI	5.12 4.87–5.38	5.68 5.43–5.93	5.96 5.69–6.23	<0.0001	6.13 5.43–6.82	6.59 5.92–7.26	7.64 6.92–8.37	0.0121
Vitamin A with supplementation (µg), mean, CI	1286.2 1126–1446	1124.0 968–1281	1199.2 1030–1369	0.3651	938.6 822–1055	1078.7 966–1191	1125.5 1004–1247	0.0854
Vitamin A with supplementation/1000 kcal (µg), mean, CI	521.0 456.4–585.6	513.4 450.1–576.6	593.0 524.5–661.6	0.1956	570.1 492.4–647.8	695.0 620.0–770.2	752.0 670.8–833.1	0.0064
Zinc with supplementation (mg), mean, CI	11.86 11.49–12.03	11.73 11.36–12.09	11.12 10.73–11.52	0.0209	8.28 8.00–8.57	8.50 8.22–8.77	8.98 8.68–9.27	0.0043
Zinc with supplementation/1000 kcal (mg), mean, CI	4.97 4.86–5.08	5.19 5.08–5.30	5.25 5.12–5.37	0.0026	4.95 4.79–5.12	5.32 5.17–5.48	5.68 5.52–5.86	<0.0001
Iron with supplementation (mg), mean, CI	13.14 12.68–13.59	12.47 12.02–12.91	12.10 11.62–12.58	0.0087	10.12 9.64–10.60	9.94 9.48–10.41	10.31 9.81–10.81	0.5722
Iron with supplementation/1000 kcal (mg), mean, CI	5.48 5.32–5.63	5.56 5.40–5.71	5.76 5.59–5.93	0.0483	6.00 5.72–6.28	6.20 5.93–6.47	6.58 6.28–6.87	0.0213
Copper with supplementation (mg), mean, CI	1.28 1.24–1.32	1.25 1.21–1.29	1.24 1.20–1.28	0.4049	1.01 0.98–1.05	1.05 1.02–1.09	1.13 1.09–1.17	<0.0001
Copper with supplementation/1000 kcal (mg), mean, CI	0.54 0.53–0.55	0.56 0.55–0.57	0.60 0.58–0.61	<0.0001	0.61 0.59–0.63	0.66 0.65–0.68	0.71 0.69–0.73	<0.0001
Manganese with supplementation (mg), mean, CI	4.72 4.55–4.89	4.78 4.61–4.95	4.60 4.42–4.78	0.3548	3.89 3.76–4.02	3.98 3.85–411	4.21 4.07–4.35	0.0035
Manganese with supplementation/1000 kcal (mg), mean, CI	2.07 1.99–2.15	2.21 2.13–2.28	2.27 2.19–2.35	0.0023	2.43 2.33–2.52	2.58 2.49–2.66	2.74 2.65–2.84	<0.0001

**Table 5 nutrients-12-00518-t005:** Main food sources of dietary total antioxidant capacity, according to socioeconomic status (SES) score.

Main Food Sources of FRAP [mmol (%)]	Men				Women			
	SES			*p*	SES			*p*
	Low (1–8)	Medium (9–18)	High (19–63)		Low (1–8)	Medium (9–18)	High (19–63)	
Coffee	3.71 (30.06%)	3.64 (28.87%)	3.74 (29.92%)	0.0533	3.82 (33.32%)	4.35 (35.14%)	4.78 (36.10%)	0.0002
Tea	2.90 (23.53%)	2.64 (20.93%)	2.41 (19.25%)	<0.0001	2.73 (23.79%)	2.40 (19.34%)	2.40 (18.14%)	<0.0001
Vegetables	2.20 (17.83%)	2.28 (18.13%)	2.17 (17.33%)	0.4891	1.83 (15.93%)	1.84 (14.86%)	1.68 (12.66%)	0.0253
Fruits	1.66 (13.44%)	1.74 (13.78%)	1.69 (13.51%)	0.5307	1.85 (16.10%)	2.19 (17.67%)	2.17 (16.36%)	0.3002
Cereals	0.60 (4.83%)	0.61 (4.83%)	0.57 (4.52%)	0.0195	0.42 (3.65%)	0.42 (3.37%)	0.42 (3.14%)	0.6230
Nuts and seeds	0.36 (2.94%)	0.59 (4.65%)	0.59 (4.70%)	0.0237	0.16 (1.38%)	0.36 (2.89%)	0.80 (6.02%)	0.0001
Chocolate and cocoa	0.22 (1.74%)	0.32 (2.52%)	0.51 (4.07%)	0.0003	0.19 (1.70%)	0.31 (2.53%)	0.41 (3.12%)	0.0004

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
