# Peer review of "Dietary Habits and Dietary Antioxidant Intake Are Related to Socioeconomic Status in Polish Adults: A Nationwide Study"

_nutrients, 2020, doi:10.3390/nu12020518_

Round 1
Reviewer 1 Report
The manuscript is interesting and well written, but there are a few issues that the authors need to consider which might make the manuscript more interesting.
Line 107 Please indicate the cut-off value of the dichotomous scoring of each component for WHO-HDI (if possible, using supplemental table). Did salt in the original WHO guidelines be excluded from the score in your study? I think that the WHO guidelines were revised in 2015.
Line 158 Why was HDI relatively low for Polish? (compared to international standards?) Why was there a discrepancy between the results of a healthy diet (not associated with SES) and antioxidant power (associated with SES)?
Line 160 Is there a traditional Polish diet? (shaped by cultural factor?) What kind of vegetables and fruits do they prefer to eat? Do they often drink wine as alcohol? Do vodka and beer have strong antioxidant power? Do they often use herbs and spices for cooking? I want to know specifically. Does it greatly affect antioxidant power? What percentage of people are eating traditional Polish diet? Is that percentage related to the level of SES?
Line 234 What kind of nuts and seeds do Polish prefer to eat? For example, walnuts, hazelnuts, chestnuts, almond, or sunflower seeds? Does it greatly affect antioxidant power?
Conclusion From your results, it can be understood that the dietary habit is better in the group with high SES. Based on this result, what kind of foods do you specifically recommend for low SES groups. What approach of dietary education is desired?
Author Response
Response to Reviewer 1 Comments and Suggestions
Thank you kindly for your review of my work. I am convinced that your valuable comments will cause to improving the quality of this manuscript. The manuscript was corrected according to the reviewer's recommendations.
Point 1: Line 107 Please indicate the cut-off value of the dichotomous scoring of each component for WHO-HDI (if possible, using supplemental table). Did salt in the original WHO guidelines be excluded from the score in your study? I think that the WHO guidelines were revised in 2015.
Response 1: According to the WHO recommendations, the following cut-off value were adopted:
- saturated fatty acids <10 % TE (total energy without energy from alcohol),
- free sugars <10% TE,
- polyunsaturated fatty acids 6-10% TE,
- protein 10-15% TE,
- cholesterol <300 mg/day
- fibre ≥25 g/day
- fruits and vegetables ≥400 g/day.
The HDI was calculated as the sum of 7 components (range 0–7) and was classified as follows: high (6–7 points), moderate (4–5 points), and low (0–3 points).
We omitted the salt component, because the amount of added salt during preparation of meals and at the table could not be evaluated precisely. However, adding salt to already seasoned dishes was reported by ¼ of Polish adults.
These informations has been added to the manuscript.
Point 2: Line 158 Why was HDI relatively low for Polish? (compared to international standards?) Why was there a discrepancy between the results of a healthy diet (not associated with SES) and antioxidant power (associated with SES)?
Response 2: Regardless of the SES group, dietary habits of the Polish adults are below the recommended standards, as we have shown in a previously published paper [16]. However, higher SES groups consumed products with higher content of polyphenols, antioxidant vitamins and minerals. These components are not included in HDI score.
These informations has been added to the manuscript.
Point 3: Line 160 Is there a traditional Polish diet? (shaped by cultural factor?) What kind of vegetables and fruits do they prefer to eat? Do they often drink wine as alcohol? Do vodka and beer have strong antioxidant power? Do they often use herbs and spices for cooking? I want to know specifically. Does it greatly affect antioxidant power? What percentage of people are eating traditional Polish diet? Is that percentage related to the level of SES?
Response 3: It was difficult to estimate the percentage of people consuming the traditional Polish diet, because indicators of this diet have not yet been developed. However, the traditional Polish diet is characterized by a high intake of red meat and animal fat, while a low intake of fish, pulses, whole grain cereal products, vegetables, seeds and nuts. Among vegetables the most popular are potatoes, cabbage and beetroot. The most popular fruits are apples. Red wine and spices, rich in antioxidants, are not typical for Polish diet. Among alcohol beverages Polish adults most frequently drink vodka and beer, which are poor in antioxidants.
Additional informations has been added to the manuscript.
Point 4: Line 234 What kind of nuts and seeds do Polish prefer to eat? For example, walnuts, hazelnuts, chestnuts, almond, or sunflower seeds? Does it greatly affect antioxidant power?
Response 4: In this study subjects reported consuming any quantity of whole nuts, namely, almonds, hazelnuts, peanuts, pistachios, or walnuts, as individual nuts or in combination. In the Polish diet, the consumption of peanuts and sunflower seeds is most common because they are the cheapest. However, our previous study have shown that walnuts had the highest antioxidant activity [22].
Additional informations has been added to the manuscript.
Point 5: Conclusion From your results, it can be understood that the dietary habit is better in the group with high SES. Based on this result, what kind of foods do you specifically recommend for low SES groups. What approach of dietary education is desired?
Response 5: The low SES groups should consume food that is relatively inexpensive, but has a high antioxidant potential, such as vegetables (red cabbage, beetroots), legumes (pea, bean) and seasonal Polish fruits (strawberries, raspberries, plums, black berries) [21].
Additional informations has been added to the manuscript.
Reviewer 2 Report
it is a good work quoting the papers with the same conclusions
Author Response
Response to Reviewer 2 Comments and Suggestions
Thank you kindly for your positive review of my manuscript.
The manuscript has undergone English language editing by MDPI and received a certificate.
